# Effects of Spermine Synthase Deficiency in Mesenchymal Stromal Cells Are Rescued by Upstream Inhibition of Ornithine Decarboxylase

**DOI:** 10.3390/ijms25052463

**Published:** 2024-02-20

**Authors:** Amin Cressman, David Morales, Zhenyang Zhang, Bryan Le, Jackson Foley, Tracy Murray-Stewart, Damian C. Genetos, Fernando A. Fierro

**Affiliations:** 1Institute for Regenerative Cures, University of California Davis, Sacramento, CA 95817, USA; acressman@ucdavis.edu (A.C.); damo@ucdavis.edu (D.M.); zhenyangzhang@csus.edu (Z.Z.); bynle@ucdavis.edu (B.L.); 2Sidney Kimmel Comprehensive Cancer Center, Johns Hopkins University, Baltimore, MD 21287, USA; jfoley13@jhmi.edu (J.F.); tmurray2@jhmi.edu (T.M.-S.); 3Department of Anatomy, Physiology, and Cell Biology, School of Veterinary Medicine, University of California Davis, Davis, CA 95616, USA; dgenetos@ucdavis.edu; 4Department of Cell Biology and Human Anatomy, University of California Davis, Sacramento, CA 95817, USA

**Keywords:** MSCs, spermine synthase, polyamines, osteogenesis, Snyder–Robinson syndrome

## Abstract

Despite the well-known relevance of polyamines to many forms of life, little is known about how polyamines regulate osteogenesis and skeletal homeostasis. Here, we report a series of in vitro studies conducted with human-bone-marrow-derived pluripotent stromal cells (MSCs). First, we show that during osteogenic differentiation, mRNA levels of most polyamine-associated enzymes are relatively constant, except for the catabolic enzyme spermidine/spermine N1-acetyltransferase 1 (SAT1), which is strongly increased at both mRNA and protein levels. As a result, the intracellular spermidine to spermine ratio is significantly reduced during the early stages of osteoblastogenesis. Supplementation of cells with exogenous spermidine or spermine decreases matrix mineralization in a dose-dependent manner. Employing N-cyclohexyl-1,3-propanediamine (CDAP) to chemically inhibit spermine synthase (SMS), the enzyme catalyzing conversion of spermidine into spermine, also suppresses mineralization. Intriguingly, this reduced mineralization is rescued with DFMO, an inhibitor of the upstream polyamine enzyme ornithine decarboxylase (ODC1). Similarly, high concentrations of CDAP cause cytoplasmic vacuolization and alter mitochondrial function, which are also reversible with the addition of DFMO. Altogether, these studies suggest that excess polyamines, especially spermidine, negatively affect hydroxyapatite synthesis of primary MSCs, whereas inhibition of polyamine synthesis with DFMO rescues most, but not all of these defects. These findings are relevant for patients with Snyder–Robinson syndrome (SRS), as the presenting skeletal defects—associated with SMS deficiency—could potentially be ameliorated by treatment with DFMO.

## 1. Introduction

Patients with Snyder–Robinson syndrome (SRS) exhibit severe skeletal defects characterized by osteoporosis early in life. Patients with SRS harbor loss-of-function mutations in the gene *Sms*, which encodes spermine synthase (SMS), an enzyme that catalyzes the conversion of spermidine into spermine (Figure 1). In consequence, an increased spermidine-to-spermine ratio is pathognomonic of SRS [1]. However, the mechanism for how this polyamine imbalance causes SRS remains unknown.

Previous reports have suggested that polyamines play an important role during osteogenesis. Rath and Reddi reported that during ectopic endochondral bone formation in rats, putrescine, spermidine, and spermine levels are initially low before spiking by the end of chondrogenesis and decreasing during ossification [2]. Other studies in rodents have suggested that both spermine and spermidine inhibit bone loss caused by ovariectomy, primarily by disrupting osteoclast activity [3,4]. However, in a large cohort of patients, sera spermidine levels are strongly correlated with increased osteoporotic fractures [5], suggesting that excess spermidine is detrimental to bone formation.

Various groups have reported on the effect of polyamines on osteogenic differentiation of MSCs in culture, with widely discordant results [6,7,8,9,10] that may be attributed to the low number of biological replicates, tissue of origin of MSCs (adipose vs. bone marrow), species, outcome measurements, and concentrations of both polyamines and inhibitors tested. Work by Albert et al. has suggested that mesenchymal stem cells/multipotent stromal cells (MSCs) derived from SRS patients show impaired osteogenic differentiation in vitro [11]. To model the disease in a dish, we have previously silenced *Sms* expression using shRNA in MSCs derived from healthy donors and confirmed that SMS deficiency impairs osteogenesis, both in vitro and after implantation into immune-deficient mice [12]. Our previous studies suggest that silencing *Sms* leads to strong dysregulation of genes and metabolites, and causes mitochondrial dysfunction [12]. However, the mechanism for how SMS deficiency impairs osteogenesis remains unknown.

Here, we report our findings on the effects of polyamines on human-bone-marrow MSCs, derived from multiple donors. We find that during osteogenesis, the polyamine catabolic enzyme spermidine/spermine N1-acetyltransferase 1 (SAT1) is increased during the first 14 days, coinciding with a sharp decrease in the spermidine-to-spermine ratio. Studies with supplementations of polyamines or inhibitors of polyamine-associated enzymes suggest that both spermidine and spermine exert a negative effect on mineralization. Chemical inhibition of SMS-impaired bone formation was rescued with difluoromethylornithine (DFMO), an inhibitor of ornithine decarboxylase (ODC1). Since DFMO has been successfully used in the clinic for other indications, it is feasible to consider this drug as a therapeutic for SRS patients. Our results warrant efficacy and safety studies in an animal model, prior to potential clinical use.

## 2. Results

### 2.1. Changes in Polyamines and Polyamine-Associated Enzymes during Osteogenesis In Vitro

To determine if polyamine levels are regulated during osteogenesis, MSCs were cultured in osteogenic media for up to 21 days. Figure 2A shows that, as expected, osteogenic markers SPP1 (osteopontin) and IBSP (bone sialoprotein) are strongly upregulated over time. Similarly, ALPL (alkaline phosphatase) increases during the early stages of osteogenesis [13]. In contrast, the expression of most polyamine-associated enzymes remained rather steady, with the remarkable exception of SAT1, which undergoes an over 10-fold increase within the first 2 weeks of differentiation. This increase in SAT1 was also observed at the protein level (Figure 2B). Since MSCs express very low levels of polyamine oxidase (PAOX; Appendix A), SAT1 is likely acting primarily as a catabolic enzyme, causing cellular secretion of both spermine and spermidine [14]. Spermidine and spermine measurements by HPLC show a significant reduction in their ratio during early osteoblastogenesis, with a possible restoration to basal levels by day 21. Putrescine was measured by mass spectrometry and found to be strongly reduced by day 7 while remaining low on days 14 and 21. The reduction in putrescine is consistent with the increase of SAT1 expression driving the export/catabolism of spermidine and spermine. These results suggest that polyamine levels are tightly regulated during osteogenic differentiation, likely by upregulation of SAT1.

### 2.2. Effect of Polyamine Supplementation on Osteogenesis

We next evaluated how exogenous polyamines may affect osteogenesis in vitro. For this, we first performed dose-response studies to assess the potential toxicity of polyamines. Putrescine was tolerated at 2.5 mM, spermidine at 1.25 mM, and spermine at 0.625 mM (Figure 3A), suggesting that in MSCs, the toxicity of polyamines is proportional to the number of amino groups. Using non-toxic polyamine levels, we found that spermine and spermidine, but not putrescine, inhibit mineralization (precipitation of hydroxyapatite) in a dose-dependent manner (Figure 3B). However, polyamine supplementation did not consistently affect alkaline phosphatase activity or gene expression of osteogenic markers (Appendix A), suggesting that these exogenous polyamines impair mineralization without directly impacting osteogenic differentiation.

Cell proliferation was not affected by putrescine or spermidine supplementation. However, spermine did increase proliferation (Figure 3C), which is consistent with our previous report using genetically modified MSCs [12]. Altogether, these results suggest that exogenous spermine and spermidine are well tolerated (at the micromolar level) but inhibit calcium apposition during osteogenesis in vitro.

### 2.3. Effect of CDAP and DFMO on Osteogenesis

To modify intracellular polyamine levels, we chemically inhibited SMS and ODC1 using CDAP and DFMO, respectively (Figure 1). CDAP is expected to lead to increased spermidine and reduced spermine, as previously described in SRS patients [11] and when using a shRNA targeting SMS [12]. Inhibition of ODC1 is expected to cause a reduction of putrescine, spermidine, and spermine [9,15]. Dose-response studies showed that CDAP becomes toxic to MSCs around 300 μM, while DFMO-treated MSCs did not show signs of toxicity at any tested concentration (up to 10 mM; Figure 4A).

Consistent with our previous results using shRNA to silence SMS [12], we found that CDAP (100 μM) inhibits cell proliferation, which could not be rescued with DFMO (20 μM; Figure 4B). These results are also consistent with our results with exogenous polyamines, suggesting that SMS deficiency leads to reduced proliferation of MSCs due to lack of spermine, and not due to accumulation of spermidine.

Chemical inhibition of SMS with CDAP impairs mineralization. Interestingly, this effect is recapitulated when CDAP is supplemented only during the first week of differentiation, but not if CDAP is supplemented on either the second or third week (Figure 4C). This robust early effect seems in line with the early changes in polyamines during differentiation (Figure 2), suggesting that impaired SMS activity causes an early effect on cells, which only later translates into impaired mineralization.

Remarkably, the inhibition caused by CDAP (200 μM) is restored by supplementation with DFMO (10 μM) (Figure 4D), suggesting that impaired SMS activity leads to reduced mineralization primarily due to the accumulation of spermidine, and not because of reduced spermine. Alkaline phosphatase activity and expression of osteogenic markers were not clearly affected by either CDAP or DFMO (Appendix A). As expected, CDAP increases the spermidine/spermine ratio. However, DFMO had little effect on restoring polyamine levels (Figure 4E). This surprising result may be related to the time exposed to the inhibitors (21 days for differentiation vs. 2 days for polyamine measurements). Of note, CDAP significantly increases acetylated spermidine (N1AcSPD), which also was not rescued with DFMO (Appendix A).

Altogether, these results suggest that cell proliferation due to SMS deficiency is not rescued by inhibition of ODC1. However, the reduced mineralization caused by CDAP can be rescued by upstream inhibition of polyamine synthesis using DFMO.

### 2.4. CDAP Induces Cytoplasmic Vacuolization, Which Is Rescued with DFMO

At high concentrations of CDAP (200 μM), MSCs show rapid cytoplasmic vacuolization (Figure 5A). These vesicles do not stain with the lipophilic dye Oil Red O or the pH-sensitive dye Acridine Orange, suggesting that these are not lipid droplets or early endosomes, respectively (not shown). Rather, ultrastructural analysis suggests that these vesicles correspond with stress vacuoles (Figure 5C), likely caused by hypoosmotic stress [16,17]. Remarkably, this strong cellular phenotype is reversed with DFMO (20 μM Figure 5B), suggesting that these stress vacuoles form due to the accumulation of spermidine. However, this effect could not be replicated by spermidine supplementation (not shown), possibly due to cellular regulation of exogenous polyamine uptake.

### 2.5. The Effect of CDAP on Mitochondria Is Partially Reversible with DFMO

Electron micrographs show that MSCs treated with CDAP exhibit large bright cristae-type mitochondria, while control MSCs show predominantly dense cristae-type mitochondria (Figure 5C). This difference in mitochondrial morphology suggests altered activity [18]. In fact, CDAP (100 μM) causes a significant increase in mitochondrial membrane potential, which is reversible with DFMO (20 μM) (Figure 6A). Similarly, Seahorse measurements showed that basal oxygen consumption rate is increased with CDAP and rescued with DFMO (Figure 6B). However, CDAP did not affect glucose consumption or expression of mitochondrial stress response genes ATF4, ATF5, and CHOP (Appendix A). Altogether, these experiments suggest that inhibition of SMS in MSCs causes mitochondrial alterations that can be reversed by upstream inhibition of ODC1.

## 3. Discussion

Polyamines are polycationic molecules and may interact with a wide range of negatively charged molecules, such as DNA, phospholipids, and specific protein domains [19]. While SRS patients likely show polyamine imbalance ubiquitously across tissues, only certain tissues are consistently affected in all patients. This organ-specific phenotype is consistent with SMS-deficient model organisms [20,21] and suggests that it is unlikely that polyamines play a major function by ubiquitously binding to negatively charged macromolecules. Rather, specific cell types could be differentially susceptible to a polyamine imbalance, based on their unique makeup. Based on the current literature, the reduced mineralization observed in MSCs undergoing osteogenesis could be attributed to alterations in the following protein functions:A.Polyamines inhibit carbonic anhydrases (CA) [22], which are necessary for bone formation [23,24]. Therefore, excess spermidine may directly impair mineralization by inhibiting carbonic anhydrases.B.Barba-Aliaga et al., recently showed that hypusinated eIF5A is required for collagen I synthesis [25]. Since spermidine is the sole precursor of hypusine, excess spermidine may cause aberrant translation of collagen I. Uncoordinated collagen I assembly is the primary cause for osteogenesis imperfecta [26], whereas SRS has been misdiagnosed as osteogenesis imperfecta due to their similarity in skeletal defects [27].C.Hypusinated eIF5A is also critical to mitochondrial function [28,29], suggesting that the impaired mineralization could be linked to our findings on mitochondrial defects. In support of this hypothesis, both mineralization and mitochondrial effects exerted by CDAP could be rescued with DFMO. However, our proliferation assays suggest that mitochondrial function was not critically compromised, because the inhibitory effect of CDAP on proliferation could not be rescued with DFMO. On the contrary, our data suggests that proliferation is reduced due to reduced spermine. Conversely, spermine supplementation promotes cell proliferation.D.Polyamines modulate specific ion channels [30,31], which play an important role in bone formation. These different mechanisms (and potentially others) for how a polyamine imbalance may affect mineralization may overlap and are not mutually exclusive.

The observed strong increase in SAT1 expression during osteogenesis and the reduction in spermidine/spermine ratio and putrescine during differentiation strongly suggest that polyamine levels are tightly regulated during osteogenic differentiation. It had been previously reported that inhibition of polyamine synthesis with DFMO promotes osteogenic differentiation of human-bone-marrow-derived MSCs and that polyamine levels are reduced during osteogenesis [9]. Our results showing that DFMO can partially rescue the effects of SMS inhibition by CDAP opens an intriguing therapeutic avenue for SRS patients. However, it is intriguing that DFMO, at the concentrations that could rescue mineralization, vacuolization, and mitochondrial defects, did not restore spermidine/spermine ratios as we had expected [32]. This could be due to DFMO inducing AMD1 activity [32], or other undetermined compensatory mechanisms. Regardless of the exact mechanism, future experiments in animals to determine the safety and efficacy of DFMO are critical, prior to clinical studies in SRS patients.

Since spermine oxidase (SMOX), the enzyme catalyzing the opposite reaction of SMS, is associated with myogenic differentiation and pathophysiology [33,34,35], it is possible that polyamine imbalances affect the differentiation of different cell types through a conserved mechanism. Although MSCs express low levels of SMOX, future experiments should examine if inhibition of SMOX [36] could reverse some of the effects caused by CDAP, presumably by restoring polyamine levels.

Inhibition of SMS with CDAP is expected to cause both, a reduction of spermine and an accumulation of spermidine. The cytoplasmic vacuolization observed with CDAP is likely associated with excess spermidine since this effect was reversible with DFMO. In fact, most known inducers of vacuolization are small amine-containing molecules [16]. Mechanistically, when these weak bases enter acidic compartments (such as late endosomes and lysosomes), they become protonated and cannot escape the compartments, causing an osmotic imbalance [37]. High concentrations of CDAP are likely to induce cell death through paraptosis, which is characterized by cytoplasmic vacuolization.

The inhibition in mineralization and mitochondrial effects (increased membrane potential and increased oxygen consumption rate) by CDAP are also likely caused by accumulated spermidine because these effects are rescued with DFMO. In line with this notion, spermidine (and spermine) supplementation also inhibited mineralization. However, spermidine supplementation did not affect mitochondrial membrane potential (not shown), perhaps because cells may be able to limit spermidine uptake during the short exposure (48 h). In contrast, the decrease in cell proliferation by CDAP is likely associated with a reduction in spermine, because this effect could not be reversed with DFMO. Supporting this concept, spermine supplementation increased cell growth.

Altogether, our studies suggest that polyamines are critically regulated during osteogenic differentiation, whereas both excess and lack of specific polyamines strongly impact MSCs at multiple levels. Further uncovering of the underlying molecular causes and tests in animals are critical for the development of a potential treatment for patients with SRS.

## 4. Materials and Methods

### 4.1. Isolation and Expansion of MSCs

MSCs were isolated from commercially available human bone marrow aspirates (StemExpress, Sacramento, CA, USA). Fresh samples were centrifuged over Ficoll at 700× *g* for 30 min. Isolated mononuclear cells were then plated directly into plastic culture flasks, using standard culture media comprised of Minimum Essential Medium alpha (MEMα, Corning, Corning, NY, USA) supplemented with 10% fetal bovine serum (FBS, Gemini, West Sacramento, CA, USA). Following 3 days in culture, cells were washed with phosphate-buffered saline (PBS, Corning, Corning, NY, USA) to discard non-adherent cells. During subsequent expansion, the media was changed every 2–3 days. MSCs in between passages 3 and 7 were used for experimentation.

### 4.2. Osteogenic Differentiation In Vitro

MSCs were seeded in triplicate in tissue-culture plates at 10,000 cells/cm^2^. The following day, the medium was changed to osteogenic media (standard culture media supplemented with 0.2 mM ascorbic acid (cat# A4544, Sigma Aldrich, St. Louis, MO, USA), 0.1 μM dexamethasone (cat# 1176007, USP, Rockville, MD, USA), and 10 mM β-glycerolphosphate (cat# 50020, Sigma Aldrich) with medium changes every 3–4 days for up to 21 days.

To measure alkaline phosphatase (ALP) activity, after 10 days in osteogenic media, cells were lifted using Trypsin, lysed with ALP lysis buffer (1.5 M Tris-HCl + 1% Triton X-100), and vortexed for 20 s. Samples were then placed in ice with strong agitation for 20 min and centrifuged at 12,000× *g* for 10 min. The supernatants were then incubated with ALP substrate *p*-nitrophenyl phosphate (pNPP, cat# P7998 Sigma Aldrich) at 37 °C for 10 min. The enzymatic product was measured by absorbance at 405 nm. ALP activity is expressed relative to the protein concentration, which was quantified using Quick Start Bradford 1x Dye Reagent (Bio-Rad, cat# 5000205, Hercules, CA, USA) and read at 595 nm.

Hydroxyapatite mineralization was measured using Alizarin Red S (ARS, Spectrum AL200) staining. Cells were cultured in either 12- or 24-well plates (ARS values vary among figures according to the surface area of the culture well) for 21 days in osteogenic media. Then, cells were fixed with 10% *v*/*v* formalin solution for 15 min, washed once with PBS, and stained with 1% *w*/*v* ARS indicator for 20 min. Following three washes with PBS, wells were documented photographically. Then, wells were incubated at room temperature with 1 mL of 10% *v*/*v* acetic acid for 30 min, scraped to remove cells, vortexed for 30 s, and centrifuged at 12,000× *g* for 15 min. The optic density of the supernatants was measured at 405 nm.

### 4.3. RNA Extraction and RT-PCR

Total RNA was extracted and collected utilizing the phenol-chloroform technique [38]. Cells in monolayers were homogenized using Tri Reagent (Zymo, Irvine, CA, USA) for 5 min. Then, chloroform was added, samples were shaken vigorously, and centrifuged at 16,000× *g* for 15 min. RNA was then precipitated by adding 100% isopropanol and centrifuged at 16,000× *g* for 10 min. RNA pellets were washed once with 75% ethanol. Once ethanol was discarded, total RNA was resuspended in nuclease-free water. Reverse transcription was performed using TaqMan Reverse Transcription Reagents (Life Technologies, Grand Island, NY, USA), and mRNA was measured by real-time PCR using TaqMan Gene Expression Master Mix (Life Technologies, Carlsbad, CA, USA) with TaqMan probes for the following genes: ALPL (Hs01029144_m1), SPP1 (Hs00959010_m1), IBSP (Hs00913377_m1), ODC1 (Hs00159739_m1), SMS (Hs04998877_g1), SRM (Hs01027696_g1), AMD1 (Hs00750876_s1), SAT1 (Hs00971739_g1), SMOX (Hs00602494_m1), PAOX (Hs00382210_m1), ATF4 (Hs00909569_g1), ATF5 (Hs01119208_m1), CHOP (Hs00358796_g1), and GAPDH (Hs02786624_g1). GAPDH was used as a housekeeping gene (internal control).

### 4.4. Western Blots

The researchers seeded 200,000 MSCs in 25-cm^2^ flasks and cultured them in osteogenic media for up to 21 days. At every given time point, the cells were lysed in ice-cold RIPA buffer (Thermo Fisher, Waltham, MA, USA, cat# 89900) containing a protease and phosphatase inhibitor cocktail (Thermo Fisher, cat# 78440). Cell lysates were mixed with 2× Laemmli sample buffer (Bio-Rad, Hercules, CA, USA, cat# 1610737) containing β-mercaptoethanol, and incubated at 95 °C for 5 min, then SDS–PAGE was performed. Proteins were transferred onto PVDF membranes and incubated overnight with primary antibodies against SAT1 (1:500, Cell Signaling, Danvers, MA, USA, cat# 61586S), b-actin (1:1000, Sigma-Aldrich) hypusine (1:20,000, Millipore, cat# ABS1064), and eIF5a (1:10,000, BD Biosciences, cat# 611976). Secondary antibodies (1:2000) were then added for 1 h at room temperature, and proteins were detected and photographed using Pierce ECL Western Blotting Chemiluminescent Substrate (Thermo Fisher, cat# 32106) and Image Lab software version 6.0 (Bio-Rad).

### 4.5. Measurement of Polyamine Levels

MSCs were cultured in osteogenic media as described above. To measure spermidine and spermine, cells were lysed with 1× SSAT breaking buffer (5 mM HEPES, pH 7.2, and 1 mM DTT) following 0, 7, 14, or 21 days of osteogenic differentiation. Samples were then frozen at −80 °C for shipment to Johns Hopkins University for analysis by HPLC. Briefly, lysates were acid precipitated with 1.2 N perchloric acid, followed by dansyl chloride labeling of the polyamine-containing supernatant. Polyamines were separated on a C18 column by HPLC as previously reported [39]. Peaks were quantified using TC software (TotalChrom version 6.3, Perkin Elmer), and values are presented relative to each other, as a spermidine to spermine ratio.

To measure putrescine, we applied a standard mass spectrometry (MS) pipeline. In brief, for the same conditions and time points as for HPLC samples, cells were collected by Trypsin-treatment and scraping, centrifuged, snap-frozen using liquid nitrogen, and stored as dry cell pellets at −80 °C. Biogenic amines were extracted using the Matyash extraction method, using MTBE, methanol, and water and creating a biphasic partition. The polar phase was then dried down to completeness and run on a Waters Premier Acquity BEH Amide column. A short 4 min liquid chromatography method was used for the separation of polar metabolites from a starting condition of 100% LCMS H_2_O with 10 mM ammonium formate and 0.125% formic acid to an end condition of 100% ACN:H_2_O 95:5 (*v*/*v*) with 10 mM ammonium formate and 0.125% formic acid. A Sciex Triple-ToF scanned from 50–1500 m/z with MS/MS collection from 40–1000 selecting from the top 5 ions per cycle. Data processing was conducted with MS-Dial using an MZ-RT list for annotations in addition to a library for MS/MS matching. Metabolomic analyses were conducted by the West Coast Metabolomics Center, at the University of California, Davis.

### 4.6. Supplementation of Polyamines or Inhibitors

Putrescine (cat# 51799, Sigma-Aldrich, St. Louis, MO, USA), spermidine (cat# 49761, Sigma-Aldrich), spermine (cat# 55513, Sigma-Aldrich), CDAP (N-cyclohexyl-1,3-propanediamine, cat# A0891, Tokyo Chemical Industry, Tokyo, Japan), and DFMO (DL-α-Difluoromethylornithine, cat# D193, Sigma Aldrich) were all added to media at concentrations shown in each figure. For studies involving administration of exogenous polyamines, media was also supplemented with 1 mM aminoguanidine (cat# 396494, Sigma Aldrich), including that of control cells, to minimize oxidative stress caused by bovine serum amine oxidase [40].

### 4.7. Dose-Response Studies

3-(4,5-Dimethyl-thiazol-2-yl)-2,5-diphenyl tetrazolium bromide (MTT) assays were conducted to assess the effects of polyamines and polyamine inhibitors on the viability of MSCs. Here, MSCs were plated into 96-well plates at 5000 cells/well in quadruplicate. The next day, cells were treated with the proper conditions and incubated for 48 h. Then, cells were washed once with PBS and incubated for 2 h at 37 °C in MSC culture media containing 5% MTT solution (5 mg/mL). Then, stop solution (Promega, cat# G4101) was added and plates were placed at 4 °C overnight. Plates were read by a spectrophotometer at 570 nm with a reference reading at 650 nm.

### 4.8. Cell Proliferation Assays

To measure cell proliferation, we used a CyQUANT Direct Cell Proliferation Assay (Thermo Fisher, cat# C35006). MSCs were plated in triplicate per condition into 96-well plates (500 cells per well). The next day, the cells were treated with the proper conditions with medium change every 3 days. At each indicated time point, a plate was washed with PBS and frozen at −80 °C. At least 24 h after freezing the last time point, plates were treated with lysis buffer and a DNA-binding fluorescent dye as described by the manufacturer. Plates were measured by fluorescence with excitation at 480 nm and emission at 520 nm.

### 4.9. Quantification and Characterization of Stress Vacuoles

MSCs were plated in triplicate at 5000 cells per well in 12-well plates and cultured for 24 h in standard culture media supplemented with inhibitors. Then, 9 different fields of view from each well were selected randomly to count cells containing vacuoles and cell nuclei (total cells).

### 4.10. Transmission Electron Microscopy (TEM)

MSCs were seeded on Permanox plastic chamber slides (cat# 12-565-20, Thermo Scientific) at 5000 cells/cm^2^. The next day, the medium change was changed to standard MSC culture media with or without CDAP, and cells were incubated for 3 days. Then, cells were fixed overnight at 4 °C using 2.5% glutaraldehyde and 2% paraformaldehyde in 0.1 M sodium phosphate buffer and submitted to the Electron Microscopy Core at UC Davis. Here, samples were dehydrated and embedded in resin. Sections were cut using an ultra-microtome and mounted on a grid. Sections were treated with heavy metals and inspected using TEM (Phillips).

### 4.11. Measurement of Mitochondrial Membrane Potential

To assess mitochondrial membrane potential, about 50,000 cells were seeded into 6-well plates. The next day, the medium was changed to the respective condition, and cells were incubated for an additional 48 h. Then, cells were treated with 1 μM orange MitoTracker dye in MEMα (without FBS) and incubated for 30 min. Then the cells were lifted with trypsin and centrifuged at 3600 rpm for 5 min. The supernatant was discarded, and the pellet was resuspended in PBS, prior to analysis using flow cytometry (Attune).

### 4.12. Bioenergetic Profiling

Oxygen consumption rate (OCR) and extracellular acidification rate (ECAR) were measured using Seahorse XFe96 (Agilent Technologies, Santa Clara, CA, USA). Cells were plated at 10,000 cells/cm^2^ 72 h in advance and pre-treated with either 100 μM CDAP, 20 μM DFMO, or both within normal culture media. Cells were lifted 24 h prior to measurement and replated at 8000 cells/well into the XFe96 well plate in quadruplicates and treated with the same conditions mentioned above. Media was then replaced with Seahorse XF DMEM medium with 4.5 g/L glucose and 15% FBS and incubated in a non-CO_2_ incubator for 1 h. Media was replaced again with fresh DMEM-XF immediately before measurement. Baseline OCR and ECAR were measured before performing an inhibitory analysis using 1 mM oligomycin, 5 mM FCCP, then 0.1 mM Rotenone, and 1 mM Antimycin A.

### 4.13. Statistical Analysis

In each figure legend, *n* indicates the number of biological replicates (MSCs derived from different donors) averaged, with error bars representing the standard error of the mean (SEM). The number of technical replicates for each individual experiment is described in the respective methods section. Statistical significances were calculated using either a 1-way ANOVA, followed by post hoc Tukey’s multiple comparison tests, or a paired Student’s *t*-test, as described in figure legends. Statistical analyses were performed using GraphPad Prism version 9.5.

## 5. Conclusions

Our studies suggest that excess polyamines, especially spermidine, negatively affect hydroxyapatite synthesis of primary MSCs, whereas inhibition of polyamine synthesis with DFMO rescues most, but not all, of these defects.

## Figures and Tables

**Figure 1 ijms-25-02463-f001:**
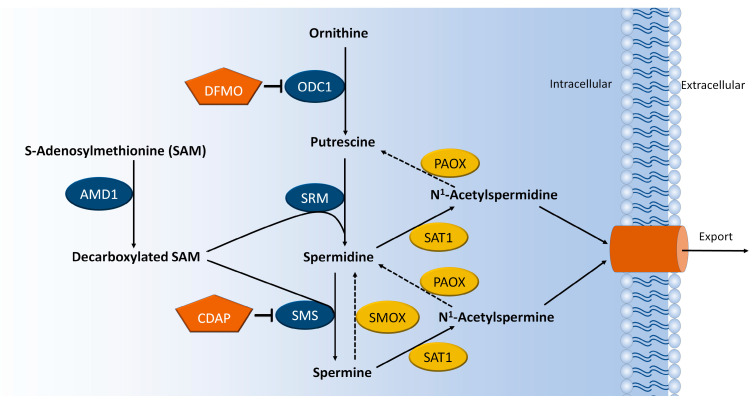
**Schematic overview of the polyamine synthesis pathway.** Abbreviations: AMD1: Adenosylmethionine decarboxylase 1, ODC1: Ornithine decarboxylase, SRM: Spermidine synthase, SMS: Spermine synthase, SMOX: Spermine oxidase, SAT1: Spermidine/spermine N1-acetyltransferase, PAOX: Polyamine oxidase, DFMO: Difluoromethylornithine, CDAP: N-cyclohexyl-1,3-propanediamine.

**Figure 2 ijms-25-02463-f002:**
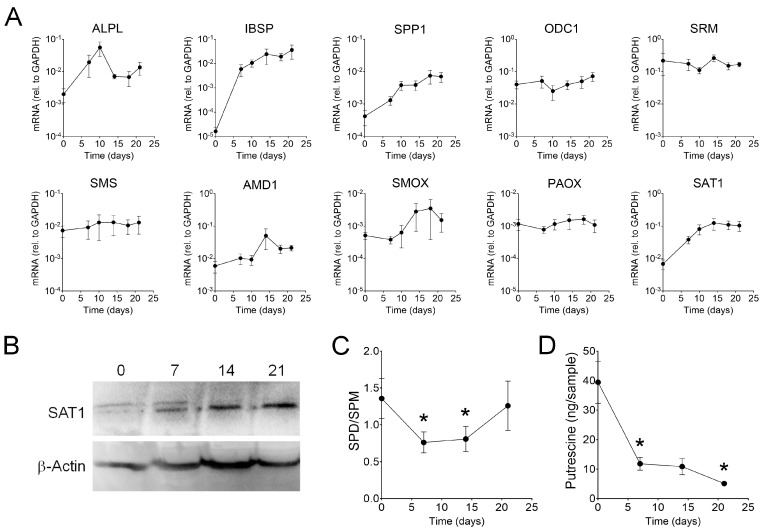
**Polyamine regulation during osteogenesis. MSCs were cultured for 21 days in osteogenic media.** (**A**) mRNA levels of osteogenic markers (ALPL, IBSP, and SPP1) and polyamine-associated enzymes (n = 5). (**B**) Representative image of a Western blot showing increased SAT1 (24 kDa) expression during differentiation (n = 3). (**C**) HPLC measurements of spermidine (SPD) and spermine (SPM) (n = 6). (**D**) Putrescine levels measured by mass spectrometry (n = 4). * *p* < 0.05 calculated using a non-paired Student’s *t*-test comparing the given time point to day 0 (baseline).

**Figure 3 ijms-25-02463-f003:**
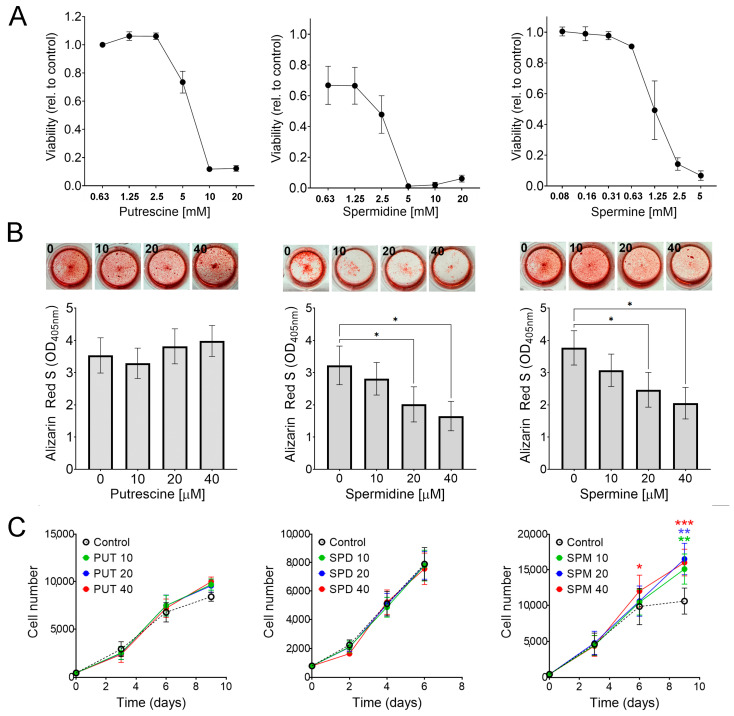
**Effect of polyamine supplementation on viability, mineralization, and proliferation of MSCs.** (**A**) Dose-response studies to test toxicity of polyamines after 48 h (n = 4). (**B**) Alizarin Red S staining of MSCs undergoing osteogenesis for 21 days in the presence of aminoguanidine and polyamines at various concentrations. Representative wells are shown above each bar graph. * *p* < 0.05 calculated by 1-way ANOVA and post hoc Tukey’s test (n = 6). (**C**) Proliferation assays on MSCs supplemented with aminoguanidine and polyamines at various concentrations (all micromolar, n = 5). * *p* < 0.05; ** *p* < 0.005, *** *p* < 0.0005 compared to control, calculated using a paired Student’s *t*-test.

**Figure 4 ijms-25-02463-f004:**
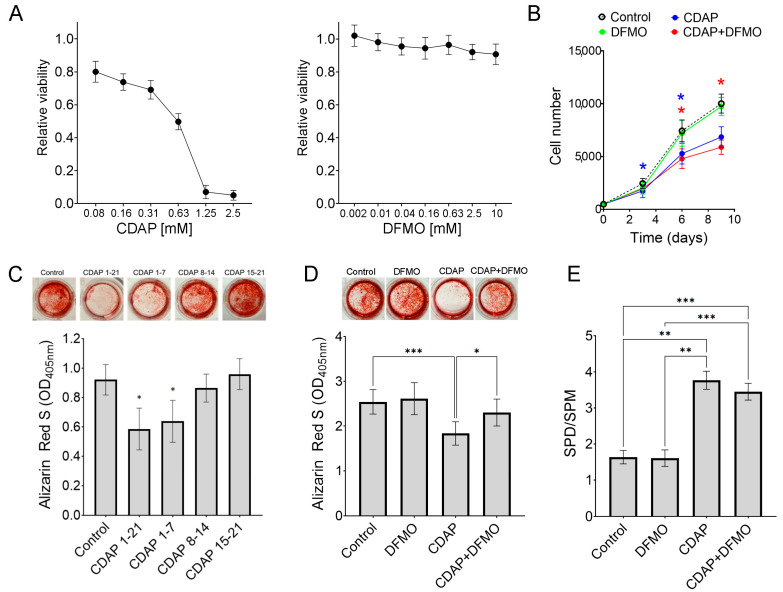
**Effect of polyamine-associated enzyme inhibitors on cell viability, mineralization, and proliferation.** (**A**) Dose-response studies of MSCs culture for 48 h with either DFMO or CDAP (n = 4). (**B**) Proliferation assays on MSCs, where CDAP is 100 μM and DFMO is 20 μM (n = 4). (**C**) Mineralization of MSCs after 21 days in osteogenic media, showing representative wells. CDAP (200 μM) was supplemented either continuously (days 1–21), or only during the first (1–7), second (8–14), or third week (15–21) (n = 7). (**D**) The mineralization reduced by CDAP (200 μM) is rescued by supplementation with DFMO (10 μM) (n = 8). (**E**) Spermidine (SPD) to spermine (SPM) ratio measured by HPLC in cells treated for 48 h with either DFMO (20 μM), CDAP (100 μM), or both (n = 4). * *p* < 0.05 ***p* < 0.005, and *** *p* < 0.0005, calculated by 1-way ANOVA and post hoc Tukey’s test (**D**,**E**) or paired Student’s *t*-test (**B**,**C**).

**Figure 5 ijms-25-02463-f005:**
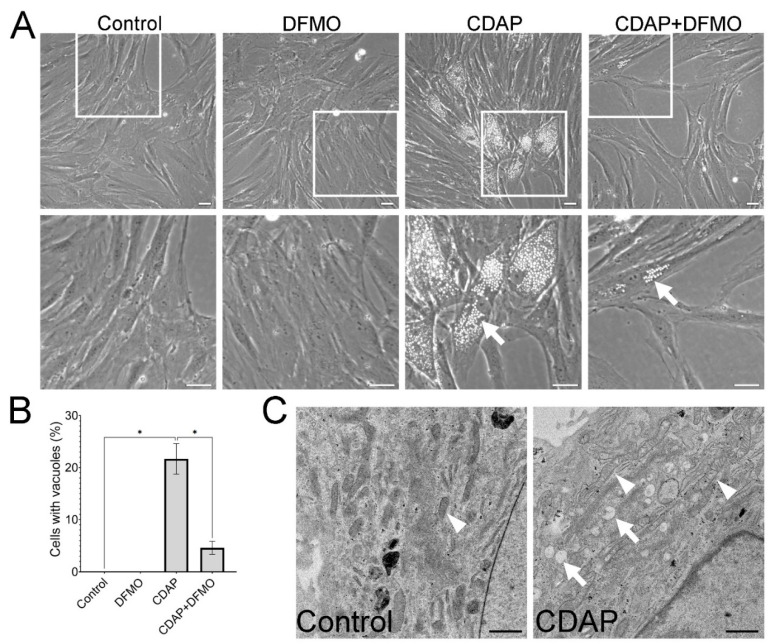
**CDAP (200 μM) causes cytoplasmic vacuolization, which is reversible with DFMO (20 μM).** (**A**) Representative phase contrast images of MSCs after 48 h with the respective inhibitors. Cytoplasmic vacuoles are shown with an arrow. The bar scale is 50 μm. (**B**) Quantification of cells with cytoplasmic vacuolization. * *p* < 0.05 calculated by 1-way ANOVA and post hoc Tukey’s test (n = 4). (**C**) Transmission electron microscopy images of MSCs treated for 48 h with or without CDAP (200 μM). Vacuoles are shown with arrows and mitochondria with arrowheads. Control MSCs show dense cristae-type mitochondria, while mitochondria in MSCs treated with CDAP look larger and with bright cristae-type mitochondria.

**Figure 6 ijms-25-02463-f006:**
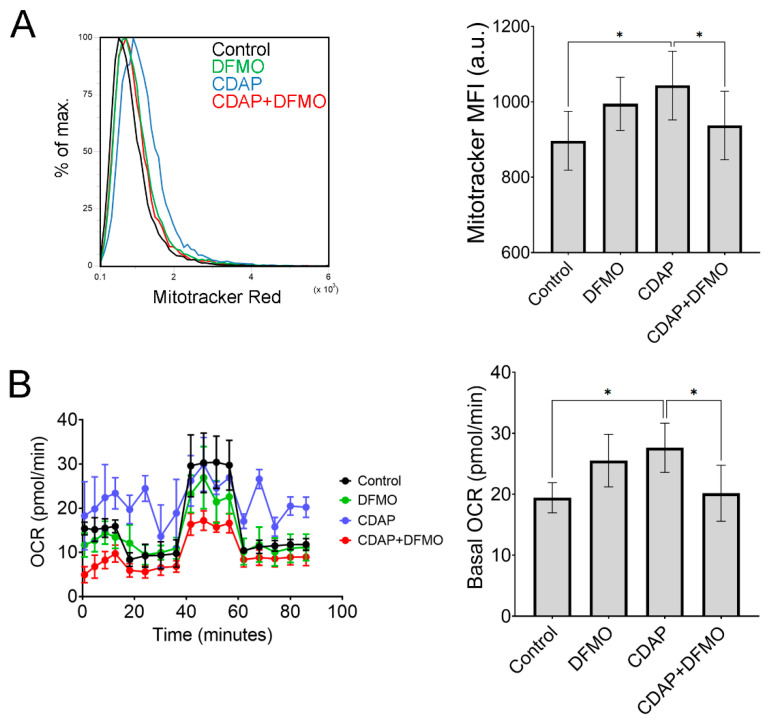
**Effects of CDAP on mitochondria are mostly reversible with DFMO.** (**A**) Mitochondrial membrane potential after exposure for 48 h to CDAP (100 μM), DFMO (20 μM), or both, measured by MitoTracker Red dye and flow cytometry. Representative histograms with quantification are shown in the bar graph. * *p* < 0.05 calculated by 1-way ANOVA and post hoc Tukey’s test (n = 6). (**B**) Representative Seahorse experiment and quantification of basal oxygen consumption rate (OCR) in MSCs treated for 48 h as above. * *p* < 0.05 calculated by 1-way ANOVA and post hoc Tukey’s test (n = 11).

## Data Availability

Data is contained within the article and Appendix A.

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
