# Peer review of "Effects of Spermine Synthase Deficiency in Mesenchymal Stromal Cells Are Rescued by Upstream Inhibition of Ornithine Decarboxylase"

_ijms, 2024, doi:10.3390/ijms25052463_

Round 1
Reviewer 1 Report
Comments and Suggestions for Authors
The manuscript submitted for publication by Fernando A. Fierro and collaborators “Effects of Spermine Synthase deficiency in Mesenchymal Stromal Cells are rescued by upstream inhibition of Ornithine Decarboxylase” investigates the role of polyamines in osteogenesis by means of a series of in vitro studies with humans bone marrow derived-multipotent stromal cells (MSCs). The conclusions drawn by the authors indicate that polyamines and spermidine in particular can slow down the synthesis of hydroxyapatite in primary MSCs. Evidence of this effect was obtained by inhibiting the synthesis of polyamines with a well-known ornithine derivative (DFMO). The subject of the manuscript is interesting and opens a new horizon on the role of polyamines in osteogenesis. However, the results appear to conflict with a previous publication which states that: “Putrescine can promote the proliferation and osteogenic differentiation of MSCs, suggesting the potential application of putrescine as a novel inductive agent for in vitro osteogenesis of MSCs [Jing-Li Chen et al. Putrescine Promotes Human Marrow Mesenchymal Stem Cells to Differentiate Along Osteogenic Pathway.2015,23(3):809-813-(doi:10.7534/j.issn.1009-2137.2015.03.040). It would be appropriate to explain in the submitted manuscript any inconsistencies (if any) with respect to the publication mentioned.
Author Response
We are thankful to the reviewer for revising our manuscript and for the positive feedback. Unfortunately, we are unable to comment on the abovementioned article, as it is only available in Chinese. The journal is Zhongguo Shi Yan Xue Ye Xue Za Zhi. Based on the abstract, we should notice that we did not test putrescine at levels as high as 100 uM. We also did not measure RUNX2 mRNA levels. From the abstract, we could not tell what was the effect on alkaline phosphatase activity and bone mineralization was apparently not measured. Based on these arguments, we respectfully request not to cite or comment on this manuscript.
Reviewer 2 Report
Comments and Suggestions for Authors
In the article entitled, “Effects of Spermine Synthase deficiency in Mesenchymal Stromal Cells are rescued by upstream inhibition of Ornithine Decarboxylase”, the authors investigated the effects of polyamines on human bone marrow MSCs derived from donors. This is a well-conducted study with potential applications in patients with Snyder-Robinson Syndrome, those presenting skeletal defects. The manuscript is written well, and the authors have employed various approaches to confirm the results.
Minor comment
1. While there are marked differences in the levels of polyamines and enzymes of polyamine metabolism during osteogenic differentiation, the authors have not mentioned the role of spermine oxidase in this context. This could be added in the discussion. Several studies have reported the role of SMOX in skeletal muscle pathophysiology.
2. The images of MSCs undergoing osteogenesis could be enlarged for better visualization.
Author Response
We are thankful to the reviewer for revising our manuscript and for the positive feedback. We have included a small paragraph discussing the potential role of SMOX. We respectfully request to leave ARS images in their current size, which has become a standard to represent this data. Our images should be at high resolution, allowing up to 500% zooming in, without image pixelation.
Reviewer 3 Report
Comments and Suggestions for Authors
The manuscript reports an experimental work that discovers new details about how polyamines regulate osteogenesis and skeletal homeostasis. Of particular interest is the application of their finding to reduce the effects of Snyder-Robinson syndrome with DFMO. Perhaps it will be beneficial for the paper if one discusses DFMO ability to cross blood-brain barrier.
Overall the paper is well written and I have no suggestions for improvement.
Author Response
Thank you for the positive feedback.
Round 2
Reviewer 1 Report
Comments and Suggestions for Authors
I thank the authors for the understandable response. In fact, the cited manuscript is not easily available. In any case, nothing detracts from the quality of their certainly interesting manuscript